# Phenotypic and Molecular Patterns of Resistance among *Campylobacter coli* and *Campylobacter jejuni* Isolates, from Pig Farms

**DOI:** 10.3390/ani11082394

**Published:** 2021-08-13

**Authors:** Dimitrios Papadopoulos, Evanthia Petridou, Konstantinos Papageorgiou, Ioannis A. Giantsis, Georgios Delis, Vangelis Economou, Ilias Frydas, Georgios Papadopoulos, Maria Hatzistylianou, Spyridon K. Kritas

**Affiliations:** 1Laboratory of Microbiology and Infectious Diseases, School of Veterinary Medicine, Aristotle University of Thessaloniki, 54124 Thessaloniki, Greece; dpapvet@hotmail.com (D.P.); pgkostas@yahoo.gr (K.P.); ilias.frydas@gmail.com (I.F.); skritas@vet.auth.gr (S.K.K.); 2Faculty of Agricultural Sciences, University of Western Macedonia, 53100 Florina, Greece; igiants@agro.auth.gr; 3Laboratory of Pharmacology, School of Veterinary Medicine, Aristotle University of Thessaloniki, 54124 Thessaloniki, Greece; delis@vet.auth.gr; 4Laboratory of Hygiene of Food of Animal Origin-Veterinary Public Health, School of Veterinary Medicine, Aristotle University of Thessaloniki, 54124 Thessaloniki, Greece; boikonom@vet.auth.gr; 5Laboratory of Animal Science, School of Veterinary Medicine, Aristotle University of Thessaloniki, 54124 Thessaloniki, Greece; geopaps@vet.auth.gr; 6Clinic of Pediatrics-Immunology and Infectious Diseases, Faculty of Health Sciences, School of Medicine, Aristotle University of Thessaloniki, 54124 Thessaloniki, Greece; mhatzistile@med.auth.gr

**Keywords:** *Campylobacter*, pigs, antimicrobial resistance

## Abstract

**Simple Summary:**

*Campylobacter* spp. has been the leading cause of human diarrhea in EU since 2005. Although poultry and poultry meat are considered as the primary source of transmission of campylobacteriosis to humans, pigs can be a significant reservoir of the pathogen, as well. Moreover, the increase of antibiotic resistance in the specific pathogen, especially against fluroquinolones and macrolides is considered a significant threat for public health. The purpose of the current study was to evaluate and molecularly characterize the antimicrobial resistance of *Campylobacter* infection in pig farms in Greece at both phenotypic and molecular level.

**Abstract:**

The purpose of this research was to characterize the antibiotic resistance patterns of *Campylobacter* spp. isolated from commercial farrow to finish farms in Greece, and analyze the relevant molecular resistance mechanisms among the resistant *Campylobacter* isolates. Susceptibility testing to five different classes of antibiotics was performed in 100 *C. coli* and 100 *C. jejuni*, previously isolated and identified. All isolates were found susceptible to meropenem. Very high rates of resistance were recorded for tetracyclines (84.5%), medium rates of resistance were recorded regarding quinolones (23%), and low and very low rates of resistance were identified for macrolides such as erythromycin and aminoglycosides (12% and 4%, respectively). Only 12.5% of the *Campylobacter* isolates displayed MDR. Regarding the molecular mechanisms of resistance, all ciprofloxacin resistant isolates hosted the mutant type *Thr-86-Ile* region of the quinolone resistance-determining region (QRDR) of the *gyrA* gene. In all erythromycin resistant isolates, the transitional mutations A2075G and A2074C in the *23S rRNA* gene were only amplified. Molecular screening of tetracycline resistance genes indicated that the vast majority of *Campylobacter* isolates (92.3%) were positive for the *tet(O)* gene. In summary, these findings and especially the very high and medium rates of resistance for tetracyclines and fluroquinolones, respectively recommend that a continuous monitoring of *Campylobacter* isolates susceptibility in combination with the proper use of antimicrobials in livestock production is of great importance for public health.

## 1. Introduction

*Campylobacter* spp. are common pathogenic bacteria of both veterinary and human public health importance. They constitute the most common human gastrointestinal pathogens reported in EU since 2005 [1]. In 2018 in EU, the number of laboratory confirmed cases of human campylobacteriosis was 246,571, corresponding to a notification rate of 64.1 per 100,000 population. Within the same year, 524 campylobacteriosis outbreaks in total have been recorded in the 28 EU Member States, 522 of which were food borne whereas the remaining two were waterborne [1]. In USA, it is estimated that 2.1–2.4 million cases of human campylobacteriosis occur every year [2]. The most common sources of *Campylobacter* transmission are raw milk and chicken meat.

Transmission occurs via the fecal-oral route after ingestion of contaminated food and water. The disease symptoms vary from a self-limiting watery diarrhea to a severe inflammatory diarrhea with abdominal pain and fever. Not infrequently, *Campylobacter* infections can be burdened with complications that can lead to chronic health problems. The main recognized sequelae after *Campylobacter* infection that can be triggered include Guillain-Barré syndrome (GBS) [3,4], reactive arthritis (REA) [5], and irritable bowel syndrome (IBS) [6].

Management of human campylobacteriosis is based on fluid therapy, which is generally considered the therapeutic corner stone. Antimicrobial treatment is only required for patients presenting more severe disease clinical signs, as well as for those who are immunocompromised. The most common antimicrobial agents implicated in the treatment of *Campylobacter* infections are macrolides, such as erythromycin, and fluoroquinolones, such as ciprofloxacin [7]. Tetracyclines have been suggested as an alternative choice for the treatment of clinical campylobacteriosis, but are rarely used in clinical practice [7].

During the past two decades, an increasing number of *Campylobacter* strains have developed resistance to fluoroquinolones and other antimicrobials such as macrolides. Moreover, the World Health Organization (WHO) identified *Campylobacter* as one of the high priority antimicrobial resistant pathogens, regarding its resistance to fluoroquinolones [7]. The resistance to both macrolides and fluoroquinolones is of major public health concern as it narrows therapeutic options for *Campylobacter* infections. Keeping this in mind, the EU continuously monitors the *Campylobacter* spp. prevalence and their resistance rates, in humans, animals, and food products. Therefore, it is considered as a public health priority.

Members of the *Campylobacter* genus exhibit optimal growth when cultured at 42 °C. They are generally isolated from the intestines of cattle, sheep, swine, and the poultry caecum. Due to a higher body temperature, poultry as well as other avian species are among the most common edible animals hosting *Campylobacter* spp., representing the main source of infection for humans [8].

Globally, *Campylobacter jejuni* is more prevalent in poultry, whereas *Campylobacter coli* is more common in pigs [9]. *C. jejuni* may co-exist with *C. coli* in pigs, but is typically detected in 10–100-fold lower levels than *C. coli* [10,11]. Pigs are considered as a natural reservoir of *Campylobacter*, exhibiting a prevalence of 50% and 100% with excretion levels ranging from 10^2^ to 10^7^ cfu/gr of feces [12,13]. In a consensus study conducted in Canada, amongst 1200 faecal samples examined, originating from 80 pig farms, 1.194 were positive for *Campylobacter* species. The prevalence of *C. coli, C. lari,* and *C. jejuni* were 99.2%, 0.6%, and 0.2%, respectively [14].

Sows have been identified as the major source for piglet contamination. Piglets are usually infected within the first days after their birth and genotypic analysis has provided evidence that sows and piglets share similar profiles [15]. In an experimental study conducted by Young et al., newborn piglets exhibited an average incidence 57.8% of *Campylobacter* within the first 24 h after birth [16]. In another study by Alter et al. [13] in 15 pig farms, none of the 1-day old newborn piglets were positive for *Campylobacter* but the average prevalence in piglets increased within the first days of life to 32.8%. In the third week of age, the prevalence of *Campylobacter* positive piglets reached 41%, increasing to 56.6% at 4 weeks of age and reaching 66.8% at 24 weeks after birth [13].

In pigs, the association of diarrhea with *Campylobacter* infection was firstly reported in 1948 by Doyle [17,18], who was able to reproduce a clinical syndrome, the so-called “pig dysentery”, in healthy pigs through experimental inoculation. Dysentery was also observed when *C. coli* was inoculated in gnotobiotic piglets by the oral route. Sala et al. [19] observed diarrhea, bacteremia, and bacterial distribution in many other organs such as lungs, kidneys, and liver of pigs after experimental inoculation with *Campylobacter.*

At the farm level, infections with *Campylobacter* are associated mainly with lactating piglets and include fever and mild to moderate diarrhea. Dehydration and loss of appetite may also occur. In sows, nursery, and fattening piglets, clinical signs are rarely observed.

The purpose of the current study was to evaluate and molecularly characterize the antimicrobial resistance of *Campylobacter* isolates from pig farms in Greece.

## 2. Materials and Methods

### 2.1. Samples and Processing

A total of 200 *Campylobacter* isolates, originating from 16 commercial pig farms from six different Greek regions were examined. The median size of the farms was 550 breeding sows (230–1600). Among these isolates, 100 have been previously identified as *C. jejuni* and 100 as *C. coli*. Speciation of *Campylobacter* isolates has been made according to a multiplex PCR protocol [20]. All samples were cryopreserved at −70 °C, in *Brucella* broth (OXOID, UK) supplemented with 5% lysed horse blood (OXOID, UK) and 15% glycerol.

The cryopreserved *Campylobacter* samples were defrosted, revived, and inoculated on to modified charcoal cefoperazone deoxycholate agar (mCCDA-OXOID, UK) with the selective supplement SR0155 (OXOID, UK). The inoculated plates were incubated under microaerobic conditions (85% N_2_, 10% CO_2_, and 5% O_2_) at 41.5 °C for 48 h.

### 2.2. DNA Isolation

Bacterial DNA for PCR was extracted using the conventional boiling method. Briefly, *C. jejuni* and *C. coli* colonies were suspended in 250 μL of ΤΕ (Tris-HCl [10 mM]: EDTA [1 mM]) buffer and were homogenized by vortexing. Suspensions were boiled at 100 °C for 10 min and were immediately placed in an iced bath for another 10 min. After centrifugation at 13,500× *g* for 10 min, the supernatant [100 μL] were collected and transferred to new tubes, and stored at −20 °C for molecular analysis to detect antibiotic-resistance genotypes by PCR.

The quantification of extracted DNA was performed spectrophotometrically and the quality of the extracted DNA was estimated from the ratio of absorbance at 260/280 nm. A value range of 1.8–2.2 was considered to indicate DNA isolation of high purity.

Antimicrobial susceptibility testing (AST).

The agar dilution method was applied for the antimicrobial susceptibility testing. All *Campylobacter* isolates were examined for their susceptibility to five antimicrobials of five different antimicrobial classes. The antimicrobials tested included gentamicin (GEN), erythromycin (ERY), ciprofloxacin (CIP), tetracycline (TET), and meropenem (MER) (Sigma-Aldrich). The Muller Hinton agar supplemented with 5% mechanically defibrinated horse blood and 20 mg/L β-NAD was used for *Campylobacter* isolates susceptibility testing.

The European Committee on Antimicrobial Susceptibility Testing (EUCAST) breakpoint tables version 11.0 for *C. jejuni* and *C. coli* were used for the interpretation of the results [21]. Regarding gentamicin and meropenem, tables of the same version for *Enterobacteriaceae* were used. Antibiotics exhibiting phenotypic resistance to more than three different classes were regarded as Multidrug Resistant [22]. The *C. jejuni* ATCC 33560 and *C. coli* ATCC 33559 were used for the AST, Quality Control.

### 2.3. Genotypic Characterization of Fluoroquinolone Resistance

All *C. jejuni* and *C. coli* isolates that were found resistant to ciprofloxacin, were examined for the presence of Thr-86 to Ile mutations (C-to-T transition) in the quinolone resistance-determining region (QRDR) of the *gyrA* gene [23,24]. Determination of the *gyrA* gene presence was performed by applying the mismatch amplification mutation assay PCR and using the FastGene Taq DNA PCR Kit (Nippon Genetics, Düren, Germany) following the manufacturer’s recommendations, with primers (Table 1) and conditions as in Zirnstein et al. [22,23].

### 2.4. Genotypic Characterization of Macrolide Resistance

The resistance in macrolides was explored by the examination of point mutations at positions 2075 and 2074 in the domain V of the 23S rRNA gene [25], and of the presence of the ribosomal RNA methylase gene, *ermB*, that was amplified as described by Qin et al. (2014) [26] using the PCR amplification kit.

### 2.5. Genotypic Characterization of Tetracycline Resistance

For the evaluation of resistance to tetracyclines, three genes, i.e., *tet(O)*, *tet(A)*, and *tet(B)* were analyzed among the *Campylobacter* isolates. PCR amplification of these genes was performed using the FastGene Taq DNA PCR Kit following the manufacturer’s recommendations, with primers (Table 1) and reaction conditions as described by Abdi-Hachesoo et al. [27].

**Table 1 animals-11-02394-t001:** Primer sequences used for species identification and detection of resistance genes and mutations.

Target Gene	Sequence (5′-3′)	Amplicon Size (bp)	Reference
*tet(O)*	F:AACTTAGGCATTCTGGCTCAC	515	[27]
	R:TCCCACTGTTCCATATCGTCA		
*tet(A)*	F: GTGAAACCCAACATACCCC	888	
	R: GAAGGCAAGCAGGATGTAG		
*tet(B)*	F: CCTTATCATGCCAGTCTTGC	774	
	R: ACTGCCGTTTTTTCGCC		
*cmeB*	F:GACGTAATGAAGGAGAGCCA	1166	[28]
	R:CTGATCCACTCCAGCTATG		
*gyrA* *Thr-86-Ile mutations* *(C. jejuni)*	F: TATGAGCGTTATTATCGGTC	265	[24]
	R: TAAGGCATCGTAAACAGCCA		
*gyrA* *Thr-86-Ile mutations* *(C. coli)*	F:TATGAGCGTTATTATCGGTC	192	[24]
	R:TAAGGCATCGTAAACAGCCA		
*23S rRNA at position* *2074*	F:TTAGCTAATGTTGCCCGTACCG	485	[25]
	R: AGTAAAGGTCCACGGGGTCTCG		
*23S rRNA at position* *2075*	F:TTAGCTAATGTTGCCCGTACCG	486	[25]
	R:TAGTAAAGGTCCACGGGGTCGC		
*ermB*	F:TGAAAAAGTACTCAACCAAAT	692	[26]
	R:TCCTCCCGTTAAATAATAGAT		

### 2.6. Genotypic Characterization of Efflux Pumps

Finally, the *cmeB* gene was analysed molecularly in all *Campylobacter* spp. strains for the presence of the multidrug efflux pumps, using primers and the PCR amplification procedure suggested by Pumbwe et al. [28].

In all analyses, PCR products were visualized by electrophoresis using an agarose gel stained with ethidium bromide.

### 2.7. Statistical Analysis

Comparisons between rates were performed after the preparation of contingency tables (chi-squared tests), as provided in the IBM^®^ SPSS^®^ version 25 statistical software (IBM Corp., Armonk, NY, USA). The level of significance was set at 5% (α = 0.05).

## 3. Results

### 3.1. Antimicrobial Susceptibility Testing (AST)

From the totally 200 examined *Campylobacter* isolates, 23 (11.5%), nine (9%) *C. coli,* and 14 (14%) *C. jejuni* isolates were found susceptible to all five antibiotic classes tested, while 25 isolates (12.5%), 15 *C. coli,* and 10 *C. jejuni* were classified as multidrug resistant by showing resistance to three different classes of antibiotics.

All clinical isolates were susceptible to meropenem. Very high rates of resistance were recorded for tetracycline, i.e., 169 *Campylobacter* isolates (84.5%), were resistant. More specifically, 88 (88%) and 81 (81%) of *C. coli* and *C. jejuni*, respectively, presented resistance in this class of antibiotics. Medium rates of resistance were recorded regarding fluroquinolones as 46 isolates (23%), 22 (22%) and 24 (24%) of *C. coli* and *C. jejuni,* respectively were resistant in ciprofloxacin. Finally, low and very low rates of resistance were identified for macrolides and aminoglycosides. For erythromycin, 24 isolates (12%), 13 (13%) and 11 (11%), *C. coli* and *C. jejuni*, respectively, were exhibiting resistance, while only eight isolates (4%), five *C. coli* and three *C. jejuni* were found resistant to gentamicin. It is noteworthy that all erythromycin resistant *Campylobacter* isolates, presented high level resistance against the selected antibiotic with MIC ≥ 32 mg/L. Statistical analysis did not reveal significant differences (*p* > 0.05) concerning the rates of resistance to any of the investigated antibiotics between *C. coli* and *C. jejuni* isolates. However, a clear pattern was discerned within both bacterial species, with resistance to ciprofloxacin being significantly (*p* < 0.001) less frequent than the resistance to tetracycline, and significantly more frequent that the resistance to erythromycin, gentamicin, and, self-evidently, meropenem.

Among all *Campylobacter* isolates, three MDR phenotypes were determined. The resistance CipEryTet phenotype was the most common, as it was present in 17 *Campylobacter* spp. isolates (8.5%). The phenotype CipGenTet followed, as it was identified in seven isolates (3.5%). In one isolate (0.5%), we observed resistance in erythromycin, gentamicin, and tetracycline. Moreover, three *Campylobacter* spp. isolates (1.5%), were resistant to both erythromycin and ciprofloxacin, drugs of choice for the treatment of invasive human campylobacteriosis. All nine resistance phenotypes recorded in the current study are shown in Table 2 and resistance rates of *Campylobacter* isolates in Table 3.

### 3.2. Antibiotic Resistance Genes (Molecular Mechanisms of Resistance)

Molecular screening of tetracycline resistance genes indicated that 92.3% of *Campylobacter* isolates (156/166) were positive for *tet(O)*. Particularly, the *tet(O)* genetic locus was detected in 94.3% of *C. coli* isolates (83/88) and 90.1% of *C. jejuni* isolates (73/81). The *tet(A)* locus was only found in 6.5% of *Campylobacter* spp. isolates (11/169). More specifically, eight *C. jejuni* and only three *C. coli* isolates were harboring the *tet(A)* gene. It should also be noted that we found two *C. coli* isolates positive for both *tet(O)* and *tet(A)* genes. None of the *Campylobacter* isolates was found positive for the *tet(B)* resistance gene.

Concerning the ciprofloxacin resistant isolates (*n* = 46), they all hosted the mutant type *Thr-86-Ile* region of the quinolone resistance-determining region (QRDR) of the *gyrA* gene.

In all erythromycin resistant isolates (*n* = 24), the transitional mutations A2075G and A2074C in the 23S rRNA gene were only amplified. The *ermB* gene was not identified in any isolate.

In a significant number of the isolates tested for the presence of efflux pumps as a resistance mechanism, the *cmeB* gene was amplified. More specifically, 23.1% of the *C. coli* (*n* = 21) and 4.6% of *C. jejuni* (*n* = 4) that were characterized as resistant in at least one of the antibiotic classes, were found to harbor the *cmeB* gene.

No significant differences were revealed between *C. coli* and *C. jejuni* in all but one of the above percentages. More specifically, the presence of the *cmeB* gene was more frequently (*p* < 0.05) amplified in the resistant *C. coli* compared with the *C. jejuni* isolates.

The frequencies of common mutations and genes conferring resistance to fluoroquinolones, tetracyclines, macrolides, and influx pumps are presented in Table 4.

## 4. Discussion

The AMR rates recorded in our study and more specifically the low and very low rates of erythromycin and gentamicin resistance, are in line with the results reported in the EFSA-ECDC summary report on antimicrobial resistance for 2017. The median EU resistance rates for tetracycline were 51.5%, and for erythromycin and gentamicin, 15.6% and 7.7%, respectively, in 979 *C. coli* isolates from fattening pigs. In regards to fluoroquinolones such as ciprofloxacin, we have determined a significant lower rate of resistance compared to 52.3% of the EFSA report [29]. Moreover, for erythromycin resistance, the low rates of resistance that we have recorded are in great difference with the data from China [30], where Tang et al. found that all 23 (100%) *C. coli* isolates from pigs and overall 75.3% of *Campylobacter* isolates from poultry and pigs were resistant to erythromycin. These differences can be attributed to the fact that macrolides, including erythromycin, are the only antibiotics authorized by the Chinese government for use as feed additives [31].

For tetracycline, the median EU resistance rate for 2017 was 51.5%. Spain has recorded in the EFSA-ECDC summary report on antimicrobial resistance for 2017, a 65.3% of resistance to tetracycline from *Campylobacter* isolates in fattening pigs. On the contrary, we recorded extremely high AMR rates for the specific antibiotic, exceeding 84% of the total isolates. Higher than the median EU rate but not as high as in our study rates, tetracycline resistance from pigs investigated in China were 64% of the *Campylobacter* isolates resistant to tetracycline [30]. In accordance with our findings, Padungtod et al. (2006) [32] reported 88% tetracylcin resistance in *Campylobacter* isolates from pigs in Northern Thailand.

The EU median for MDR *C. coli* according to the EFSA report [29] was 21.2%. We recorded that MDR *Campylobacter* spp. isolates were relatively lower (12.5%), in particular 15% for *C. coli* and 10% for *C. jejuni.* The most common MDR phenotypes identified in the current study (CipEryTet and CipGmTet) are those recognized by the EFSA report.

According to the third joint inter-agency report on integrated analysis of antimicrobial agent consumption and occurrence of antimicrobial resistance in bacteria from humans and food-producing animals in the EU/EEA, titled “Antimicrobial consumption and resistance in bacteria from humans and animals” (ECDC, EFSA, EMA, 2021), consumption of tetracyclines in food-producing animals in Greece for the year 2018 (48.9 mg/PCU) far exceeded the mean value of the 31 countries included (31.7 mg/PCU), ranking sixth overall [33]. The extensive use of tetracyclines in livestock production may be related with the high rates of resistance.

On the contrary, consumption of macrolides (4.1 mg/PCU) significantly lagged behind the European mean value (8.0 mg/PCU), whereas consumption of both fluoroquinolones and aminoglycosides (2.2 and 6.5 mg/PCU, respectively) was very close to the overall means (2.5 and 6.4 mg/PCU, respectively) [33]. It is noted that meropenem is not authorized for use in food-producing animals, at least within the European Union, and since it belongs to Category A (Avoid), according to the EMA/CVMP/CHMP classification (2019), its use is exceptionally allowed in companion animals only. Antibiotic consumption data that indicate limited to moderate use of macrolides, aminoglycosides, and fluroquinolones in livestock production could explain the moderate and low resistance rates in the specific classes of antibiotics. Moreover, the susceptibility of all *Campylobacter* isolates to meropenem is related with the fact that it is not authorized for use in livestock production.

The principle molecular mechanism for ciprofloxacin resistance of *Campylobacter* is the alteration of codon 86 from threonine to isoleucine in the *gyrA* genomic region [34]. All the ciprofloxacin phenotypically resistant *Campylobacter* isolates in our study shared the same mechanism. Our findings are similar to Woźniak-Biel et al. (2016) [35] and El-Adawy et al. (2012) [36], who revealed the same mutation in all ciprofloxacin-resistant *Campylobacter* strains from broilers and turkeys, respectively. Moreover, Tang et al. [30] characterized the T86I amino acid substitution as the sole mutation recorded in quinolones resistant *Campylobacter* isolates from poultry and pigs.

Concerning the macrolides molecular mechanism of action for resistance, we did not detect the presence of the *ermB* gene in any of the isolates tested. Erythromycin resistance in all *Campylobacter* isolates was determined by detecting point mutations at position 2075 and 2074 in the V 23S rRNA gene. The high resistance levels observed in our study (MIC ≥ 32 mg/L) are in agreement with the correlation of the specific resistance mechanism with high resistance levels [37]. On the other hand, our results are in divergence with those revealed by Tang et al. [30], where 52.7% of *Campylobacter* isolates, mainly from poultry, were found positive for the *ermB* gene. Furthermore, more than half of the *ermB*-positive isolates also demonstrated the A2075G 23S rRNA mutation. Only three *Campylobacter* isolates from pigs were found to carry the *ermB* gene.

In 92.3% of all *Campylobacter* isolates (*n* = 156), the specific *tet(O)* gene was amplified providing evidence that resistance against tetracycline was mediated mainly through the gene, whereas in only 6.5% (*n* = 11) of all isolates, the *tet(A)* gene was identified. These results are in accordance with several previous studies [27,30,38,39] referring to the *tet(O)* gene as the principal mechanism of tetracycline resistance in *Campylobacter* isolates from different sources (animal, human, food).

The *cmeB* efflux pump is not only related with fluoroquinolone resistance, but with resistance to multiple antibiotics (macrolides, chloramphenicol, tetracycline), dyes (acridine orange), and disinfectants, as well [28,40,41]. The *cmeB* gene was amplified in 14.1% (*n* = 25) of all *Campylobacter* isolates. These findings are similar to those reported previously from *Campylobacter* isolated from turkeys [34]. Significant differences were recorded between the two *Campylobacter* species concerning the presence of the *cmeB* gene. The specific gene was amplified in 21 *C. coli* (23.1%) isolates and only in four *C. jejuni* isolates (4.7%).

## 5. Conclusions

This study was designed to assess the phenotypic and molecular patterns of resistance of *C. coli* and *C. jejuni* isolates from commercial farrow-finisher pig farms in Greece.

Fluoroquinolones and macrolides have been classified as category I and category II antimicrobials, respectively and are characterized as “first line” antibiotics for campylobacteriosis treatment. The high rates of resistance in tetracyclines and the moderate rates of fluoroquinolones resistance highlight the necessity for a continuous and systematic monitoring and surveillance of *Campylobacter* isolates from pig farms, regarding their phenotypic and molecular resistance patterns. Monitoring *Campylobacter* isolates susceptibility and the proper use of antimicrobials in livestock production are considered of great importance in order to tackle antimicrobial resistance and the spread of antimicrobial resistance pathogens and resistance genes.

## Figures and Tables

**Table 2 animals-11-02394-t002:** Antimicrobial resistance phenotypes.

Resistance Phenotypes	*C. Coli* (*n*)	*C. Jejuni* (*n*)	*Campylobacter* Spp., (*n*)	%
CipEryTet	10	7	17	8.5
CipGenTet	4	3	1	3.5
EryGenTet	1	0	1	0.5
CipEry	1	2	3	1.5
CipTet	5	9	14	7
EryTet	1	2	3	1.5
Cip	2	3	5	2.5
Tet	67	60	127	63.5
Susceptible to all antibiotics	9	14	23	11.5
Total	100	100	200	100

**Table 3 animals-11-02394-t003:** Antimicrobial resistance of *Campylobacter coli*, *Campylobacter jejuni*, and *Campylobacter* spp.

Antibiotic	*n*, %	*C. Coli* *	*C. Jejuni* *	*Campylobacter* Spp.
Ciprofloxacin	*n*	22	24	46
	%	22 ^b^	24 ^b^	23
Erythromycin	*n*	13	11	24
	%	13	11	12
Tetracycline	*n*	88	81	169
	%	88 ^a^	81 ^a^	84.5
Gentamicin	*n*	5	3	8
	%	5	3	4
Meropenem	*n*	0	0	0
	%	0	0	0

*: No differences in the resistance rates to each antibiotic were observed between C. coli and C. jejuni isolates. ^a^: Significantly higher resistance rate, compared with all other investigated antimicrobials. ^b^: Significantly higher resistance rate, compared with erythromycin, gentamicin, and meropenem; significantly lower resistance rate, compared with tetracycline.

**Table 4 animals-11-02394-t004:** Frequency of common mutations and genes conferring resistance to fluoroquinolones, tetracycline, macrolides, and efflux pumps in *C. jejuni* and *C. coli*.

Antibiotics	*n*, %	*Fluoroquinolones*	*Macrolides*		*Tetracyclines*			*Efflux Pumps*
Genes/mutations		*Thr-86-Ile*	*A2075G &* *A2074C*	*ermB*	*tet(O)*	*tet(A)*	*tet(O)& tet(A)*	*cmeB*
*C. coli*	*n*	22	13	0	83	3	2	21
	%	100	100	0	94.3	3.4	2.3	23.1 *
*C. jejuni*	*n*	24	11	0	73	8	0	4
	%	100	100	0	90.1	9.9	0	4.7 *
*Campylobacter* spp.	*n*	46	24	0	156	11	2	25
	%	100	100	0	92.3	6.5	1.2	14.1%

* Significant difference in the detection frequency between *C. coli* and *C. jejuni* isolates.

## Data Availability

Not applicable.

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
