# Peer review of "Phenotypic and Molecular Patterns of Resistance among Campylobacter coli and Campylobacter jejuni Isolates, from Pig Farms"

_animals, 2021, doi:10.3390/ani11082394_

Round 1
Reviewer 1 Report
Nice and interesting piece of research. Well done.
Author Response
We would like to thank the reviewer for her/his valuable time, kind words and interest to our research
Reviewer 2 Report
This is an interesting article, which contains some useful and important data regarding Campylobacter in pigs. This is an important area for public health and also for animal and human health. The data is robust, but appears to link to a previous study maybe which if published should be referenced. There are a few areas which need additional data adding in from that study. There are also a number of grammatical issues within the paper, and I have tried to correct those below. I hope that the comments are useful.
Please ensure that all decimal values have a full stop in them and not a comma throughout the manuscript
Line 20- perhaps in the EU may sound better?
Line 48- perhaps in the EU may sound better?
Line 50- maybe within the same year may sound better?
Line 54- maybe chicken meat may be more beneficial here?
Line 57- comma after infrequently
Line 59-61- maybe this may read better as ‘include Guillain-Barré syndrome (GBS) [3,4], reactive arthritis (REA) [5] and irritable bowel syndrome (IBS) [6].’
Line 64- comma after signs
Line 70- this needs a reference
Line 71-72- as one of the high priority antimicrobial resistant pathogens (reword)
Line 75- continuously monitors (reword)
Line 77- maybe would read better as ‘when cultured at 42°C’ ?
Line 85- comma after Campylobacter
Line 85- exhibiting a prevalence of (reword)
Line 87- faecal samples may sound better than faeces samples?
Line 93-94- average incidence 53.8% of Campylobacter …. (reword)
Line 95- none of the one day old newborn piglets were positive for (reword)
Line 97- comma after age
Line 97- the prevalence of …(reword)
Line 97 – 98 – increasing to 56.6% at four weeks of age and reaching 66.8% at 24 weeks after birth…. (reword)
Line 100- who was able to reproduce a clinical syndrome,… (reword)
Line 102- can be shorted to C. coli here
Line 102- by the oral route
Line 105- At the farm level, infections (reword and add in comma)
Line 109- Campylobacter isolates from pig farms (reword maybe)
Line 109- can delete ‘at both phenotypic and molecular level’ as repetitive
Line 113-115- I would like to see more data in here- about the big farms and how they were confirmed as these species
Line 115- Brucella should be italicised
Line 117- samples were defrosted, revived and inoculated on to ….. (reword)
Line 119- please include details of the microaerophilic conditions
Line 124- were boiled at 100C (reword)
Line 125- placed in an iced bath (reword)
Line 126- I would put 100ml in brackets here. And ml? sure its not ul?
Line 126- transferred to new tubes
Line 133- is this not more commonly known as the kirby Bauer diffusion method?
Line 135- included or were?
Line 137-139- please include antibiotic concentrations
Line 141- Enterobacteriaceae needs to be in italics
Line 144- whats AST quality control? Perhaps as a known susceptible strain?
Line 146- resistant to ciprofloxacin (reword)
Line 146-151- please break this into a few smaller sentences.
Line 156- what PCR amplification kit? Details would be good here
Line 158- why were these 3 genes chosen rather than all the other Tet resistance genes?
Line 164- strains for the presence of the …. (reword)
Line 166- reword the end bit here as it doesn’t make sense
Line 168- comma after analyses
Line 168- visualised by electrophoresis using an agarose gel … (reword)
Line 171- what do you mean by rates? Prevalence?
Line 178- delete totally
Line 178- comma after isolates
Line 178- this doesn’t make sense as there are too many numbers. What does the 23% (11.5%) refer to?
Line 178- having the % in for coli and jejuni would be useful
And in line 180.
Line 181- resistance to three different …. (reword)
Line 184- comma after specifically
Line 184- % in here may be useful too
Line 186- And % here too
Line 188-189- a % in here too may be useful
Line 190- resistant to gentamycin (reword)
Line 198- comma after isolates
Line 199- CipEryTet needs defining
Line 199- comma after common
Line 199-200- what was the breakdown of these isolates?
Also in lines 200-204
Line 218- 83 + 73 doesn’t = 159?
Line 219- comma after specifically
Line 220- It should also be noted that … (reword)
Line 228- as a resistance mechanism (reword)
Line 230- space between C and jejuni
Line 234- identified or amplified may sound better than confirmed
Line 243-244- this needs a reference
Line 245- comma after ciprofloxacin
Line 246- comma after resistance
Line 247- contrast rather than great resistance
Line 248- please use a full stop in the % rather than a comma
Line 249- differences rather than difference
Line 253- delete of
Line 254- resistance to tetracyclin (reword)
Line 257- from pigs investigated in China (reword)
Line 257- resistant in tetracyclin (reword)
Line 258- reported 88% tetracylcin resistance in Campylobacter isolates from pigs in Northern Thailand (reword).
Line 260 …according to the EFSA report …. (Reword)
Line 261- delete comma after that
Line 293- of the ermB gene …. (Reword)
Line 295-297- please reword as this is unclear.
Line 298- comma after isolates and poultry
Line 299- ermB should be in italics
Line 302- comma after n=156),
Line 304- comma after isolates
Line 308- but with resistance to multiple ….. (reword)
Line 320- remove comma from after antimicrobials, and place it after respectively.
Line 326- spread of antimicrobial resistant pathogens and resistance genes (reword)
Author Response
Please see the attachment

This manuscript is a resubmission of an earlier submission. The following is a list of the peer review reports and author responses from that submission.
Round 1
Reviewer 1 Report
Congratulations on an interesting piece of research.
Couple of suggestions:
- When the authors state their purpose, did they mean "resilience" or "resistance"? (end of section 1)
- The way the results are presented in the abstract, I would not be fully convinced of the importance of the continuous monitoring of resistance and use; but I would be, by the way they are presented in the discussion and conclusions - just a matter of editing them
- the EUCAST guidelines were used; as the authors are certainly aware, a big problem/issue is the absence of harmonisation with the CLSI ones. It would be very interesting and useful to make a comparison between them, if possible and doable
- Main gap in my opinion - making the connection with use data. What is the use data of tetracyclines in Greece? What about for the other antimicrobial classes analysed? What are the related policies in place? Have they changed in the last years? Even if it will based on their own opinions or speculations, adding these pieces would bring a lot of extra value to the publication.
Reviewer 2 Report
Dear Authors
Your manuscript could give a practical support to knowledge of Campylobacter antibiotic resistance in pigs but it presents serious flaws as the lack of statistical analyses, essential for your work . Further your discussion is poor . No considerations or hypotheses are attempted to justify your results . You reported only data , comparing them, at times, with the results obtained by other authors. Several errors are over the text (a lot of these due to a poor accuracy) . A careful revisions of English should be also opportune. I encourage you to re-write the article and to perform statistical analyses as the objectives may be of support to knowledge of antibioti-resistance epidemiology in pigs . In attachment you can find some suggestions , if you consider to resubmit the article in this journal

Reviewer 3 Report
The manuscript deals with significant information about antimicrobial resistance.
All names of genes should be in italics - ex. lines 199, 202, 212
All names of Campylobacter bacteria should be in italics - ex. lines 172, 174, 178
line 134 - please add the abbreviations of tested antimicrobials, because in Table 2 authors present resistance pattern using abbreviations
lines 83 and 85 - different ways of writing the references
line 239 - Woźniak-Biel, the mistake in the surname of an author